# A Three-Stage Dynamic Assessment Framework for Industrial Control System Security Based on a Method of W-HMM

**DOI:** 10.3390/s22072593

**Published:** 2022-03-28

**Authors:** Xudong Ji, Hongxing Wei, Youdong Chen, Xiao-Fang Ji, Guo Wu

**Affiliations:** School of Mechanical Engineering and Automation, Beihang University, Beijing 100191, China; jixd@buaa.edu.cn (X.J.); weihongxing@buaa.edu.cn (H.W.); xiaofangji163@163.com (X.-F.J.); wuguobeijing@buaa.edu.cn (G.W.)

**Keywords:** security, dynamic assessment, industrial control systems, weighted hidden Markov model

## Abstract

Industrial control systems (ICS) are applied in many fields. Due to the development of cloud computing, artificial intelligence, and big data analysis inducing more cyberattacks, ICS always suffers from the risks. If the risks occur during system operations, corporate capital is endangered. It is crucial to assess the security of ICS dynamically. This paper proposes a dynamic assessment framework for industrial control system security (DAF-ICSS) based on machine learning and takes an industrial robot system as an example. The framework conducts security assessment from qualitative and quantitative perspectives, combining three assessment phases: static identification, dynamic monitoring, and security assessment. During the evaluation, we propose a weighted Hidden Markov Model (W-HMM) to dynamically establish the system’s security model with the algorithm of Baum–Welch. To verify the effectiveness of DAF-ICSS, we have compared it with two assessment methods to assess industrial robot security. The comparison result shows that the proposed DAF-ICSS can provide a more accurate assessment. The assessment reflects the system’s security state in a timely and intuitive manner. In addition, it can be used to analyze the security impact caused by the unknown types of ICS attacks since it infers the security state based on the explicit state of the system.

## 1. Introduction

The industrial control system can be remotely interacted with and communicated with cloud services [1], cyber-physical systems [2], or edge devices in a highly networked environment [3]. Cyber-attacks are increasingly becoming a threat to ICS; thus, their security is critical [4,5]. Once an essential piece of equipment experiences a safety incident, it causes a shutdown of the system and even causes casualties [6].

The security assessment of industrial control systems is an integral part of security [7,8]. The system’s security integrates internal attributes and external environments. The elements involved in assessment have the following characteristics: large number, strong correlation, and poor accessibility. Thus, evaluating the security of the system accurately is difficult.

In recent years, security assessment research has mainly focused on artificial intelligence [9], medical subjects [10,11], infrastructure [12], power systems [13], coal mining [14], chemicals [15], etc. The industrial field focuses on assessing the system’s functional safety or the static information security assessment for the design, and its security status is easily observerd [16,17]. However, the critical equipment executing complex control tasks in ICS is challenging for capturing security statuses directly. System security information is also related to running variables, such as system operating status and environmental status. Security assessment must have the ability to obtain security information of complex systems dynamically.

New cyberattacks for which its forms and types tend to be unknown are hard to be detected [18]. Due to the limited resources of devices and networks in ICS [19,20], it is hard to obtain security information of assessments directly by detecting attacks [21]. Cyberattacks make ICS more undependable and unsafe when using the Internet [22]. Scholars generally evaluate system security through the consequences of information attacks [23]. Consequence refers to the property losses caused by an information attack. For example, Muhammad Adil et al. identified a jamming attack channel by detecting different transmission frequencies and Round Trip Time (RTT) of transmitting a signal from multi-channel in WSNs transmission media [24].

There are three methods of assessment based on consequences of attacks [20,25]: qualitative assessment, quantitative assessment [26], and the combination of them. There are many qualitative evaluation methods, such as attack trees and fuzzy calculations. These methods are coarse-grained assessments of system parameters. For example, Xu Hui et al. used attack trees to identify various attacks for security management of SDN [27]. However, they could not provide a quantitative value to evaluate the consequences of the attack. The quantitative evaluation methods can assess the impacts of information attacks. Wenli Shang et al. provided a security assessment method based on an attack tree model with fuzzy set theory and probability risk assessment technology [28]. Jingjing Hu et al. proposed a multi-dimensional network security risk assessment framework [29], including two stages: risk identification and risk calculation. They used HMM to assess the network security risk in the risk calculation stage. The HMM assessment method can effectively reflect and quantify the security risks of the physical network system. However, they did not assign the weight to the result in the risk value calculation, as the network servers and nodes have different importance. We should weigh different parts of ICS in ICS due to its heterogeneity. Nary Subramanian et al. proposed a quantitative method of NFR (non-functional requirements) safety assessment for the infrastructure system of oil pipeline systems [30]. This method can solve the integrated assessment of functional safety and security. However, it cannot calculate the assessed value of the security. Aziz A. et al. used ontology knowledge to analyze the causal relationship between events [31], established corresponding probability models, and identified the consequences of abnormalities. This method can quantitatively assess the consequences of the system’s attacks. However, the probability established by this method was stationary, while the system risk is changing. It is challenging to apply dynamic evaluations.

Currently, the most commonly used security assessment method is a combination of qualitative and quantitative information [32,33]. One is the analytic hierarchy process, which is a multi-level weight decision analysis method. Jun Chen applied the analytical hierarchy process for industrial control system evaluation [34]. Moreover, it can effectively evaluate industrial control risks. However, it cannot dynamically assess the system of ICS due to unknown attacks and threats that follow ICS. Some pieces of research had brought focus onto the necessity of a framework for the evaluation of IoT device security [35,36]. The above research methods are not dynamic and cannot meet this assessment requirement.

We propose a practical security assessment framework, a three-stage dynamic assessment framework for ICS based on a method of W-HMM. The main contributions of this work are as follows:i.The proposed method combines a qualitative and quantitative assessment of ICS security dynamically by using a W-HMM model. The method can infer the system’s risk value, which can be used as a system risk reference in a timely and intuitive manner through the explicit consequences of the attack on the device.ii.The assessment of the industrial robot control system (IRCS) is used as an example to illustrate the use of the method and compared with two typical security assessment methods.

The article is structured as follows. In the next section, we introduce the static recognition of DAF-ICSS. Dynamic monitoring is described in Section 3. Section 4 shows the assessment. Section 5 explains the framework of DAF-ICSS. Section 6 uses an IRCS as an example to verify DAF-ICSS. We discuss the results in Section 7. Finally, Section 8 summarizes the work of this paper.

## 2. Static Recognition

### 2.1. Basic Value

The ICS is a part of the company’s fixed assets, and its security will affect the value of the company’s fixed assets. We use the analytic hierarchy process to evaluate the basic value of ICS. The basic value, $B_{v}$, is used to assess the economic value of ICS. The basic value is divided into three layers: target layer, factor layer, and index layer, shown in Figure 1.

The target layer obtains $B_{v}$. The factor layer decomposes the basic value of the ICS into two critical factors: asset value and asset status. Asset value represents the economic value of the ICS, and asset status reflects the state and the environment. The index layer decomposes the factors into the fine-grained index. The asset value includes three values: self-value $S_{v}$, indirect value $A_{v}$, and accident value $A_{c}$. Self-value refers to the asset value of the ICS. Indirect value is the indirect economic loss of the enterprise caused by ICS failure without injury. accident value represents the estimated financial loss of an injury accident caused by ICS attacks. It is obtained from accident probability $f_{1}$ and accident loss *g*.
(1)$$A_{c}=f_{1}\times g$$

Asset status is divided into three states: self-state *h*, network state $N_{s}$, and work environment $E_{s}$, as shown in Table 1. The self-state reflects the performance and stability of the system, thereby affecting asset value. The latter two reflect the harshness of the system’s external environment and affect the system’s vulnerability value. Network state refers to the value determined by the network bandwidth, traffic, and peak value. The working environment is the value determined by temperature, humidity, and electromagnetic interference. The smaller the valuation, the lower the risk. The basic value is calculated as follows.
(2)$$B_{v}=S_{v}\times h+A_{v}+A_{c}$$

### 2.2. Vulnerability

The vulnerability $V_{v}$ of ICS refers to the system’s weakness that attackers can abstract [37]. When the vulnerability of an ICS is attacked, a basic attack path model should be shown in Figure 2. The attack is achieved through three steps: network path connection, data manipulation, and breaking through protection. The first step ensures that the attacker can connect to ICS through the network. The second step is that the attacker sends malicious attack instructions when the attacker could imitate external communication data of ICS. The third step hides or floods attack instructions so that the instructions can pass through the protection. The attack instructions could steal, change, or delete the system’s data. Moreover, the attack may cause the system’s faults.

The vulnerability of ICS can be illustrated from three factors depending on the three steps of the attack path model. The three factors are availability $A_{a}$, data weakness $S_{r}$, and safety protection $S_{w}$. Availability means the degree that ICS can achieve specific operations through the network when random attacks are launched. Data weakness evaluates the possibility of communication data being attacked. Security protection is the system’s ability to prevent information attacks. These three factors respectively assess the vulnerability of the three steps of the attack model. Figure 3 shows the hierarchy diagram of vulnerabilities of ICS.
(3)$$A_{a}=l\times m\times r$$
(4)$$S_{r}=log_{2}\left(2^{v}+2^{u}+2^{w}\right)$$
(5)$$S_{w}=C_{p}\times S_{p}\times F_{r}$$

The availability is divided into three indexes to measure the difficulty of attackers connecting to the system: Vector *l*, complexity *m*, and authentication *r*. *l* is used to measure the network distance between the attacker and ICS. Before an attacker can connect to the system, he must transfer instructions through network nodes such as routing equipment. The more nodes are, the more inefficiently the instructions connect. *m* describes the level of attack method that an attacker can achieve. When the attacker is connected to the system, *r* will be an index to stop the connection. The description and corresponding valuation, which are given by experts of cybersecurity, are shown in Table 2.

The data weakness is divided into three indexes: confidentiality *u*, integrity *w*, and usability *v*, as shown in Table 3.

Safety protection $S_{w}$ includes three indexes: code patch $C_{p}$, normal protective measure $F_{r}$, and emergency protective measure $S_{p}$, shown in Table 4. Code patch reflects the extent of the patches covering system vulnerabilities. Normal protective measure refers to the ability to protect the system against information attacks under normal operating conditions. The emergency protective measure is the capability to handle emergencies when in danger. The vulnerability value $V_{v}$ is shown as follows.
(6)$$V_{v}=N_{s}\times \left(2\times A_{a}+S_{r}+S_{w}\right)/3$$

## 3. Dynamic Monitoring

### 3.1. W-HMM Establishment

The HMM model can be used to build a dynamic evaluation model. Describing the stochastic process of generating explicit state sequences from hidden state sequences, HMM is a probability model related to time series. Each hidden state generates an explicit state. The security status of ICS is mostly unobservable. All the operations and the faults caused by the attack are recorded in the system’s log. The security is related to the fault with a certain observation probability. The probability of mutual transition between security states is the occurrence probability. By using the occurrence probability of the security state, the current system risk probability can be calculated to monitor the system’s risk dynamically.

However, HMM cannot distinguish the magnitude of the danger caused by different states. This paper proposes a W-HMM (weight HMM) method. W-HMM is the optimization method of HMM and weighs the results calculated by HMM. The value of weight is estimated based on the magnitude of the danger. The aim is to improve the accuracy of the evaluation when calculating the risk value. In W-HMM, we optimize the calculation of HMM results by weighing the security state. The W-HMM model is proposed, shown in Figure 4.

We construct a mapping relation of the Markov process with parameters. The security state can be categorized into secure $S_{1}$, monitored $S_{2}$, attacked $S_{3}$, and captured $S_{4}$ states. $S_{2}$ indicates that the system is scanned or spied by an attacker. In this state, the bandwidth resources are occupied, and parameters will be stolen. In the attacked state, the attacker sends malicious data, but the system has not been captured. In the captured state, the system is captured by the attacker to execute the attacker’s instructions. In this state, the system may crash or perform dangerous operations. According to the severity of the fault, system faults are classified into normal $O_{1}$, error $O_{2}$, mild alarm $O_{3}$, and warning $O_{4}$; moderate alarm $O_{5}$; and serious alarm $O_{6}$, as shown in Figure 5 and Table 5.

### 3.2. Calculating Occurrence Probability of Security by W-HMM

The specific steps are as follows.

Constructing State and Model

The explicit state set *O* and the hidden state set *S* are, respectively, shown as follows.
(7)$$O=O_{j}\left(1\leq j\leq 6\right)$$
(8)$$S=S_{i}\left(1\leq i\leq 4\right)$$

The development relationships between hidden states is related by $a_{im}\left(1\leq i,m\leq 4\right)$. $a_{im}$ is the probability of transition from state $S_{i}$ at time *t* to state $S_{m}$ at time t+1.

The hidden state is represented by the explicit state. $b_{ij}\left(1\leq i\leq 4,1\leq j\leq 6\right)$ is called the explicit state probability matrix, and shows the relationship between hidden and explicit state. $b_{ij}$ is the probability of transition from the state $S_{i}$ at time *t* to state $O_{j}$ at the time t+1.

The state transition probability matrix *A* and the explicit state probability matrix *B* can be written as follows.
(9)$$\begin{array}{cc} A & =\left[\begin{array}{cccc} a_{11} & a_{12} & a_{13} & a_{14}\\ a_{21} & a_{22} & a_{23} & a_{24}\\ a_{31} & a_{32} & a_{33} & a_{34}\\ a_{41} & a_{42} & a_{43} & a_{44} \end{array}\right] \end{array}$$
$a_{im}=P\left(x_{t+1}=S_{m}|x_{t}=S_{i}\right),\;1\leq m\leq 4,1\leq i\leq 4\;$
(10)$$\begin{array}{cc} B & =\left[\begin{array}{cccccc} b_{11} & b_{12} & b_{13} & b_{14} & b_{15} & b_{16}\\ b_{21} & b_{22} & b_{23} & b_{24} & b_{25} & b_{26}\\ b_{31} & b_{32} & b_{33} & b_{34} & b_{35} & b_{36}\\ b_{41} & b_{42} & b_{43} & b_{44} & b_{45} & b_{46} \end{array}\right] \end{array}$$
$b_{ij}=P\left(x_{t+1}=O_{j}|x_{t}=S_{i}\right),\;1\leq j\leq 6,1\leq i\leq 4$

The W-HMM of the ICS can be described as λ, among which π represents the probability of the initial state, which is shown as follows.
(11)$$\begin{array}{c} \lambda =\left(A,B,\pi \right)\\ where\;\pi =P\left(x_{i}=S_{i}\right),\;\;1\leq i\leq 4\; \end{array}$$

2.Algorithm of Baum–Welch

Markov model correction algorithms based on state sequences are classified into supervised and unsupervised learning algorithms [38]. A supervised learning algorithm records a large amount of state data to estimate the parameters. However, it is time consuming, costly, and causes difficulty in evaluating parameters dynamically. Unsupervised learning algorithms identify model parameters based on training samples and are suitable for calculating the parameters of W-HMM of ICS. To accurately describe the system and adapt to system changes, the W-HMM model is trained and updated iteratively with the Baum–Welch algorithm [39]. It is possible to obtain (see Appendix A) Equation (Equation 12), which represents the W-HMM model after n+1 iterations. When adding new sample data, the current Markov parameters are taken as the initial parameters, and Baum–Welch iterative calculations are carried out to obtain the latest parameters. The occurrence probability of security risks is extracted from the state transition probability matrix of parameters.
(12)$$\lambda^{n+1}=\left(A^{n+1},B^{n+1},\pi^{n+1}\right)$$

## 4. Assessment

Field experts of ICS obtained the state weight values that affect the risk value of ICS, shown in Table 6. The risk value describes the loss caused by an attack on the company. It is equal to the product of the failure probability and consequence of the attack. The security risk value can be determined as follows.
(13)$$\begin{array}{cc} \; & SV=E_{s}\times B_{v}\times e^{V_{v}}\times \\ \; & \left\{L_{1}\times \sum_{n=2}^{n=4}\left(\pi^{n+1}\left(i\right)\times b_{i2}\right)+\cdots +L_{5}\times \sum_{n=2}^{n=4}\left(\pi^{n+1}\left(i\right)\times b_{i6}\right)\right\} \end{array}$$

From Equation (Equation 13), $L_{1},L_{2},L_{3},L_{4}$, and $L_{5}$ are the weight values of the five states in the observation states $(O_{2},O_{3},O_{4},O_{5}$, and $O_{6})$. Since some model parameters change with time, they are classified according to the measurement of their period, demonstrated in Table 7.

## 5. Framework of DFA-ICSS

Illustrated in Figure 6, the DAF-ICSS framework is composed of static identification, dynamic monitoring, and evaluation. The framework can evaluate the security of ICS qualitatively and quantitatively. In this section, we will briefly summarize the procedure of the assessment.

During data collection, there are two stages. One is static identification, which identifies the value and vulnerability of each part of the evaluation system. The analytic hierarchy process enumerates the factors that affect the value and the vulnerability of the system. The result of static identification is represented by a severity value, which is obtained by multiplying the value and vulnerability of the system.

The other is dynamic monitoring, which calculates the system’s security risk probability. Moreover, the calculation is based on W-HMM. Its result predicts the possibility of system security risks.

The system security state is unobservable. W-HMM is introduced for dynamic monitoring to establish the connection between the observable states and the unobservable security state. This method calculates the risk probability according to the observation state and updates the risk probability in the next new observation state. W-HMM can quantify and weigh the risk probability based on different application scenarios and models.

In the assessment stage, we obtain the system risk value by multiplying the severity value and the risk probability. We develop a risk map. The map determines the risk level by the risk value’s boundary. The boundary is set by the risk tolerance of the system, shown in Figure 7. The system is safe when the risk value is in the green area. If it is in the yellow area, the system is at risk. The evaluation system will immediately issue an alarm if it reaches the red zone. Its purpose is to locate risk levels according to the calculated risk value quickly. The corresponding protection strategy can be chosen rapidly according to the level and risk value.

## 6. Experiments and Results

### 6.1. Experimental Setup

An attack test platform is built to verify the feasibility and effectiveness of the evaluation method proposed in this paper, shown in Figure 8.

The platform comprises three computers and a robot that are connected through a gigabit router. The robot, which is ER3, is produced by Effort. The controller of the robot is Robox produced by Robox SPA.

The computers play as the remote terminal, the evaluation terminal, and the attack terminal. The remote terminal sends instructions and programs. The attack terminal attacks the control system. Meanwhile, the evaluation terminal records the robot’s status data and evaluates the system.

### 6.2. Experimental Data Collection and Calculation

Step 1: Static identification

The company’s asset data and data evaluated by asset management are shown in Table 8. Table 9 displays the vector and complexity in availability. When the robot connects to the PC, the robot does not perform authentications or check the content of the communication. As a result, the authentication score is 0.704. Most system security measures are warnings. The valuation of data weakness can be found in Table 9. Most system faults are alarming. When a serious failure occurs, the system will stop running. The score of the emergency protective measure is 0.5.

Step 2: Dynamic monitoring

The initial parameters of the model are obtained by collecting and sorting empirical data. The initial parameters $A^{0}$, $B^{0}$, and $\pi^{0}$ are obtained by empirical estimation. We used two types of DOS attacks during the experiment in the two periods. In the first period, the hacker uses ping flooding random attacks. In the second period, the hacker uses UDP attacks to attack the vulnerable spots of the system. In the experiment, the attacked ports are random, and the attack frequency is once every 10 min.

Step 3: Assessment

The failure of the industrial control system caused by the attack is probabilistic. We divide the experiment into two stages. Each attack stage lasts a week, and the robot control system status is collected once an hour during the working time every day. We record the status of the control system with three different methods: DAF-ICSS, expert [40] and HMM [41]. The data obtained by these methods are shown in Table 10. We obtain the average value of each stage evaluation from the on-site enterprise expert group.
(14)$$\begin{array}{cc} A^{0} & =\left[\begin{array}{cccc} a_{11} & a_{12} & a_{13} & a_{14}\\ a_{21} & a_{22} & a_{23} & a_{24}\\ a_{31} & a_{32} & a_{33} & a_{34}\\ a_{41} & a_{42} & a_{43} & a_{44} \end{array}\right]\\ \; & =\left[\begin{array}{cccc} 0.53 & 0.26 & 0.16 & 0.05\\ 0.36 & 0.50 & 0.08 & 0.06\\ 0.11 & 0.21 & 0.52 & 0.16\\ 0.03 & 0.07 & 0.18 & 0.72 \end{array}\right] \end{array}$$
(15)$$\begin{array}{cc} B^{0} & =\left[\begin{array}{cccccc} b_{11} & b_{12} & b_{13} & b_{14} & b_{15} & b_{16}\\ b_{21} & b_{22} & b_{23} & b_{24} & b_{25} & b_{26}\\ b_{31} & b_{32} & b_{33} & b_{34} & b_{35} & b_{36}\\ b_{41} & b_{42} & b_{43} & b_{44} & b_{45} & b_{46} \end{array}\right]\\ \; & =\left[\begin{array}{cccccc} 0.72 & 0.11 & 0.08 & 0.04 & 0.03 & 0.02\\ 0.60 & 0.14 & 0.11 & 0.10 & 0.03 & 0.02\\ 0.03 & 0.01 & 0.06 & 0.3 & 0.2 & 0.4\\ 0.07 & 0.05 & 0.09 & 0.3 & 0.13 & 0.36 \end{array}\right] \end{array}$$
(16)$$\pi^{0}=\left(0.7,0.1,0.1,0.1\right)$$

Figure 9 shows that the risk value evaluated by experts is close to the other algorithms at the beginning of stage 1. It means that every assessment method is accurate at the beginning. After the beginning stage, the expert’s assessment remains the same at each stage and cannot dynamically evaluate the system’s security. HMM and DAF-ICSS assessment methods can dynamically assess the security of the system.

At the beginning of stage 2 in the assessment, we open the UDP port of the system, which leads to a new vulnerability. We increase the weight of *V_v_*, and it results in a severe alarm state. However, the HMM assessment method draws an insensitive rise in Figure 9. Because the HMM method assigns no risk weight to a variety of system security states, DAF-ICSS can sensitively reflect the changes of system risk by weighing the evaluation elements and establishing a dynamic evaluation model. When the system is attacked by a UDP flood, the risk value assessed by DAF-ICSS exceeds CNY 40,000. The values given by other evaluation methods do not exceed CNY 40,000. DAF-ICSS assessment is more accurate than HMM.

Then, the evaluation system issues an alarm and closes the UDP port to lower the risk’s value. The experiment confirms the effectiveness and timeliness of DAF-ICSS.

ICS may expose different vulnerabilities during running. For example, the operator opens the remote service port by mistake, which could lead to system vulnerability. We should focus on the dynamic changes in risk value. When the risk value suddenly increases, the system is probably experiencing a security risk.

The proposed framework of DAF-ICSS is more sensitive than other methods [42,43], which could provide a dynamical risk value. Table 11 lists the comparision of some security assessment methods in performance.

## 7. Disscusion

In addition to industrial robots, the proposed method can also be used for other industrial control systems. Some parameters in the method need to be evaluated based on a specific application scenario. For example, the work environment will be changed according to different scenarios. If we obtain a more accurate risk value, the sequence and state can be increased to enhance the quality of DAF-ICSS. It will also increase the computational burden of the system.

The security assessment combines the qualitative method, which uses a risk map to determine the system’s security, and the quantitative method, which uses a risk value to measure security value. Because the assessment relies on observations of ICS anomaly alarms, the risk value may be inaccurate when the alarms do not accurately match the system’s state. In the future, it can be overcome to some extent.

## 8. Conclusions

Security assessment is the critical part of the system’s security. We propose a security assessment method for ICS. We divide the security assessment of ICS into three steps: static recognition, dynamic monitoring, and assessment. A hierarchical system is provided for evaluating security risks. To obtain the system’s risk level, the assessment method based on W-HMM calculates the industrial security risk value. It can be updated online for optimized estimation results and determine the degree of influence with different parameters in the factory. DAF-ICSS enables operators to find out a change of risk with high precision and efficiency. It also can be used to conduct cause analysis and security impact analysis.

However, this assessment method still has room for improvement, such as exploring methods for selecting more reasonable weights, etc. In addition, there is a well-known problem in industrial systems: The already designed secure architecture that does not sacrifice functionality has difficulty in providing a coordination of security and safety. In the future, we could focus on assessing multiple industrial control systems and fundamental understanding between safety and security based on dynamic security assessments so that dedicated modeling constructs and metrics can be proposed.

## Figures and Tables

**Figure 1 sensors-22-02593-f001:**
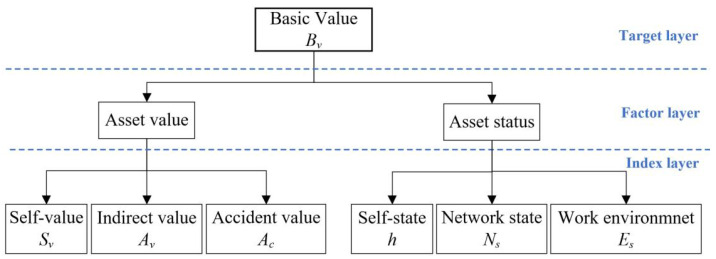
The basic value decomposition diagram.

**Figure 2 sensors-22-02593-f002:**
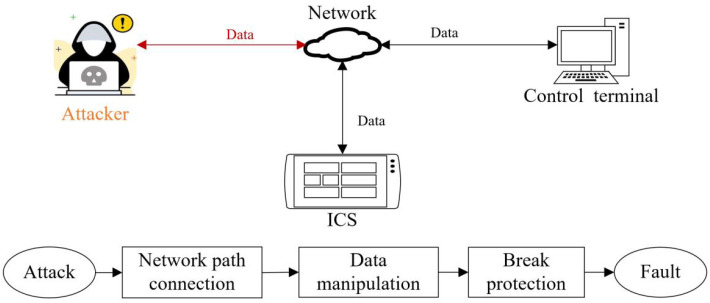
The processes of the attack path model.

**Figure 3 sensors-22-02593-f003:**
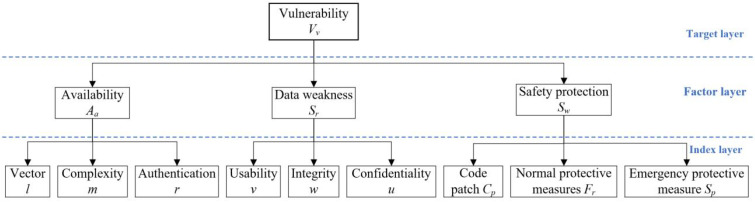
Hierarchy diagram of the vulnerability of ICS.

**Figure 4 sensors-22-02593-f004:**
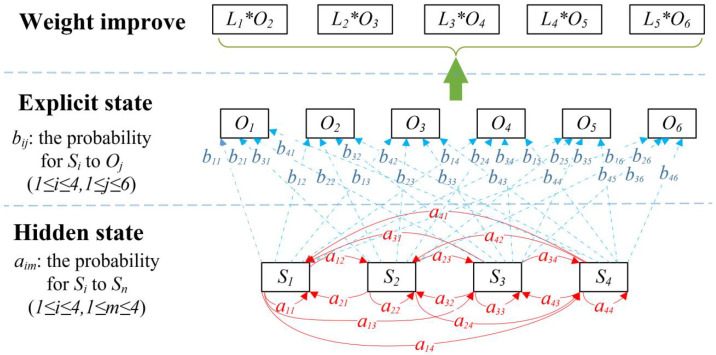
The W-HMM model.

**Figure 5 sensors-22-02593-f005:**
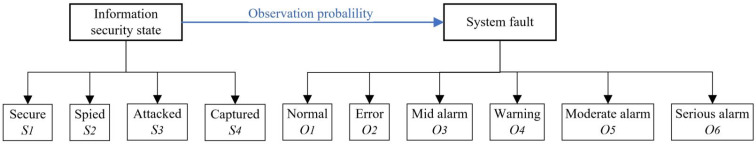
The analysis diagram of security and system fault.

**Figure 6 sensors-22-02593-f006:**
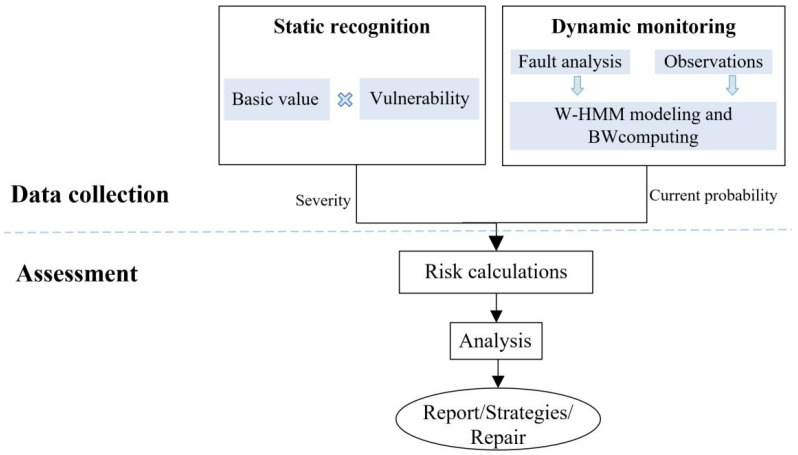
Security evaluation framework of DAF-ICSS.

**Figure 7 sensors-22-02593-f007:**
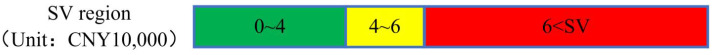
Risk map.

**Figure 8 sensors-22-02593-f008:**
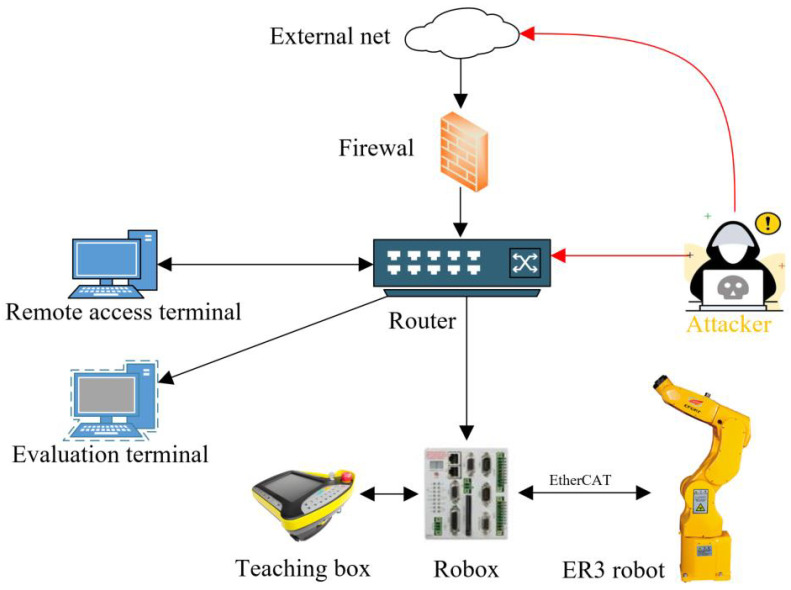
The topography of the system.

**Figure 9 sensors-22-02593-f009:**
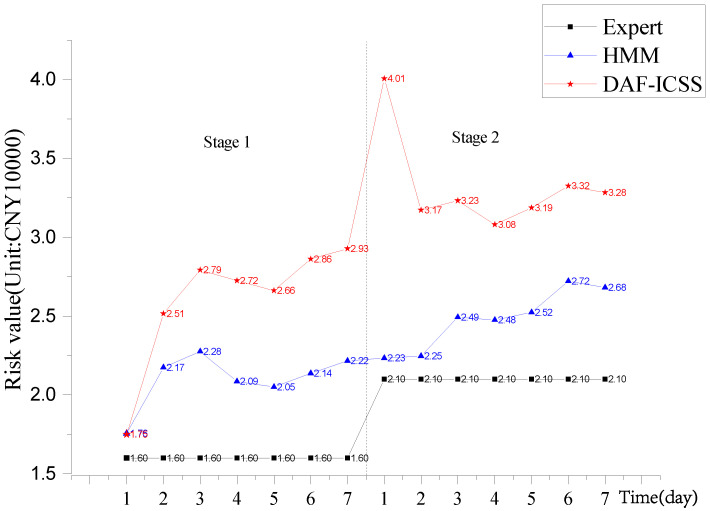
The result of the experiment.

**Table 1 sensors-22-02593-t001:** Quantitative table of asset status.

State	Description	Valuation
Self-state *h*	No-fault, available	0.25
	Fault-fixed, warning	0.5
	Fault but not affecting the main function, dangerous	0.8
Network state $N_{s}$	Network bandwidth utilization ≤50%, steady flow	0.5
	Network bandwidth utilization ≤80%, flow fluctuation	1
	Network bandwidth utilization ≥80%, flow fluctuates greatly	1.5
Work environment $E_{s}$	Temperature normal, Humidity drying, Weak electromagnetic interference	0.5
	One item is out of rating	0.8

**Table 2 sensors-22-02593-t002:** The valuation of availability.

	Description	Valuation
Vector *l*	Remote	0.85
	Neighbor	0.62
	Local	0.55
	Port physical connection	0.2
Complexity *m*	Primary	0.71
	Secondary	0.61
	Senior	0.35
Authentication *r*	Repeatedly	0.45
	Single	0.56
	None	0.704

**Table 3 sensors-22-02593-t003:** Data weakness valuation value.

	Description	Valuation
Usability *v*	Process parameters viewing commands	0.2
	System parameters viewing commands	0.3
	All parameters viewing commands	0.4
	Process parameters editing commands	0.6
	System parameters editing commands	0.8
	All parameters editing commands	1
Integrity *w*	Syntax verification audit	0.7
	Pre and post content verification audit	0.5
	Hazard verification audit	0.3
Confidentiality *u*	Encryption	0.3
	Unencrypted, nonstandard	0.5
	Unencrypted, standard	0.9

**Table 4 sensors-22-02593-t004:** Safety protection value.

	Description	Valuation
Code patch $C_{p}$	All	0.1
	Part	0.4
	None	0.7
Normal protective measure $F_{r}$	More than two	0.2
	One or two	0.6
	None	0.9
Emergency protective measure $S_{p}$	Soft response (without damaging the equipment under the premise of safety)	0.3
	Hard reaction (equipment May be damaged when ensuring safety)	0.5
	None	0.8

**Table 5 sensors-22-02593-t005:** Example of system faults.

System Faults	Identifier	Example
Normal	$O_{1}$	/
Error	$O_{2}$	Program syntax error, user password error, etc.
Mild alarm	$O_{3}$	The planning path may exceed The limit of the system, etc.
Warning	$O_{4}$	System acceleration approaching the setting threshold, etc.
Moderate alarm	$O_{5}$	Speed exceeds the threshold during running, then alarm and stop running, etc.
Serious alarm	$O_{6}$	The system detects motor overcurrent, then alarm and emergency stop, etc.

**Table 6 sensors-22-02593-t006:** State weight value.

	Weight
$L_{1}$	0.5
$L_{2}$	1
$L_{3}$	2
$L_{4}$	2.5
$L_{5}$	4

**Table 7 sensors-22-02593-t007:** Type of parameters.

Type	Identifier	Description
Constant	$L_{1}-L_{5}$	It only needs to be collected once and can be used for a long time
	$V_{v}$	
	$A_{c}$	
	$A_{v}$	
Stage constant	$E_{s}$	Regular collection and evaluation are required
	$S_{v}$	
	*h*	
Real-time volume	SV	Real-time acquisition and calculation
	π	
	*B*	

**Table 8 sensors-22-02593-t008:** Basic value datasheet of the robot unit.

Property	Name	Identifier	Valuation	Remarks
Self-value $S_{v}$	Controller, sensors and accessories, etc.	/	CNY 40,000	Collection of financial information
Indirect value $A_{v}$	Labor, equipment, product lost, etc,	/	CNY 160,000	
Accident value $A_{c}$	Accident probability	$f_{1}$	0.01	Statistics
	Accident loss	*g*	CNY 1,000,000	
Asset status	Self-state	*h*	0.25	Query the above related assessment form after evaluation
	Network state	$N_{s}$	1.5	
	Work environment	$E_{s}$	0.8	

**Table 9 sensors-22-02593-t009:** Control system information sheet.

Property	Name	Identifier	Valuation	Remarks
Availability $A_{a}$	Vector	*l*	0.55	Check the evaluation form according to the information
	Complexity	*m*	0.61	
	Authentication	*r*	0.704	
Data weakness $S_{r}$	Usability	*v*	0.5	
	Integrity	*w*	0.7	
	Confidentiality	*u*	0.5	
Safety protection $S_{w}$	Code patch	$C_{p}$	0.7	
	Normal protective measure	$F_{r}$	0.9	
	Emergency protective measure	$S_{p}$	0.5	

**Table 10 sensors-22-02593-t010:** The data of the observation sequence.

Stage 1				
Day	Observation sequence	Expert (CNY $10^{4}$)	HMM (CNY $10^{4}$)	DAF-ICSS (CNY $10^{4}$)
Day 1	$O_{1},O_{1},O_{1},O_{1},O_{1},O_{2},O_{1},O_{4}$		1.75649	1.7498
Day 2	$O_{1},O_{1},O_{1},O_{3},O_{1},O_{3},O_{2},O_{4}$		2.17419	2.51482
Day 3	$O_{1},O_{3},O_{1},O_{2},O_{1},O_{4},O_{1},O_{5}$		2.27618	2.79187
Day 4	$O_{1},O_{1},O_{1},O_{2},O_{1},O_{5},O_{1},O_{5}$	1.60	2.0854	2.72481
Day 5	$O_{1},O_{1},O_{1},O_{1},O_{2},O_{3},O_{2},O_{6}$		2.04967	2.66117
Day 6	$O_{1},O_{2},O_{1},O_{3},O_{1},O_{6},O_{2},O_{4}$		2.13676	2.86088
Day 7	$O_{2},O_{1},O_{3},O_{1},O_{4},O_{4},O_{5},O_{6}$		2.2158	2.92729
**Stage 2**				
Day 1	$O_{1},O_{5},O_{2},O_{6},O_{2},O_{4},O_{4},O_{5}$		2.23394	4.0062
Day 2	$O_{1},O_{1},O_{2},O_{3},O_{3},O_{4},O_{4},O_{5}$		2.24704	3.17175
Day 3	$O_{2},O_{1},O_{2},O_{3},O_{3},O_{2},O_{2},O_{5}$		2.4923	3.23129
Day 4	$O_{1},O_{2},O_{2},O_{1},O_{3},O_{1},O_{2},O_{6}$	2.10	2.47547	3.08055
Day 5	$O_{1},O_{3},O_{2},O_{1},O_{3},O_{5},O_{2},O_{4}$		2.52337	3.1867
Day 6	$O_{2},O_{2},O_{3},O_{2},O_{3},O_{4},O_{3},O_{4}$		2.72174	3.32432
Day 7	$O_{1},O_{2},O_{1},O_{3},O_{3},O_{5},O_{3},O_{6}$		2.68091	3.28233

**Table 11 sensors-22-02593-t011:** The comparison of some security assessment methods in performance.

Method	Qualitative or Quantitative	Accuracy	Static or Dynamic	Evaluate Unknown Attacks
Expert	Qualitative	High accuracy	Static	Y
Fault tree	Qualitative	Medium accuracy	Static	N
Bayesian network	Quantitative	Medium accuracy	Static	N
HMM	Combination	Medium accuracy	Dynamic	Y
DAF-ICSS	Combination	High accuracy	Dynamic	Y

## Data Availability

Not applicable.

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
