# Peer review of "A Three-Stage Dynamic Assessment Framework for Industrial Control System Security Based on a Method of W-HMM"

_sensors, 2022, doi:10.3390/s22072593_

Round 1

Reviewer 1 Report

In this paper, authors propose a dynamic assessment framework for industrial control system security (DAF-ICSS) based on machine learning, and take industrial robot system as an example. Paper is clearly written, logically organized, very comprehensive and impressive amount of research and study has been carried out. The content is technically sound and contains sufficient interest.

The reviewer has some concerns.

  • Some sentences are too long to make readers confused, and there are also some typos and grammar errors in this paper.
  • The quality of the figures should be improved.
  • Please improve the reference format and verify the number of each reference cited in the paper.

Author Response

Dear reviewer,

Thank you for your professional comments concerning our manuscript. Those comments are valuable and helpful for revising and improving our paper and the critical guiding significance to our research. We have studied comments carefully and have made a correction which we hope meet with approval. Revised portions are marked in red on the paper. The responses to the comments are attached.

Best regards,

Xudong Ji et al.

Reviewer 2 Report

This paper addressed the issue of cyber attacks on Industrial control systems (ICS) through proposing a dynamic assessment framework for industrial control system security (DAF-ICSS) based on machine learning, and take industrial robot system.

The paper performed security assessment from both qualitative and quantitative perspectives, combining three assessment phases which are static identification, dynamic monitoring, and security assessment.

The paper is interesting but I have some comments
1. Abstract section: the authors need to add the results and the contribution of the study.
2. The authors need to add more recent studies in cybersecurity attacks such as (1) An anonymous channel categorization scheme of edge nodes to detect jamming attacks in wireless sensor networks, (2) An energy proficient load balancing routing scheme for wireless sensor networks to maximize their lifespan in an operational environment, (3) An efficient load balancing scheme of energy gauge nodes to maximize the lifespan of constraint oriented networks, (4) Cybersecurity in Industrial Control System (ICS).
3. The authors need to discuss the results and compare with other previous studies.
My recommendation: accept after conducting the comments.

Author Response

(The authors gave the same response as above.)

Reviewer 3 Report

The article deals with a topical issue and appropriately expands knowledge. 
The title corresponds to the article.
The abstract needs to focus more on the specific outcomes of the article and its potential benefits. Conversely, I recommend omitting the description of the procedure. 
I agree with the statement in line 41, but it would be helpful to cite the literature supporting this division. 
The article does not have the usual structure of a scientific paper, but this fact does not detract from its contribution. However, I would recommend inserting at least partly a section on methodology to interpret the experiments in the future. 
I recommend that some of the calculations and matrices be included in an appendix.
In conclusion, I appreciate the mention of room for improvement and the outline of future directions. 
On the other hand, I miss the comparison of the results with similar topics or with other solutions. 
Overall, I rate the article as applicable for practice, but its contribution to science is more difficult due to the absence of the usual parts of the article. 
There are several typos and minor inaccuracies in the text.

Author Response

(The authors gave the same response as above.)

Reviewer 4 Report

Objective: a dynamic assessment framework for industrial control system security (DAF-ICSS) based on machine learning is proposed.

Methodology: the security assessment of ICS is divided into three steps: static recognition, dynamic monitoring and assessment. Security assessment is checked from both qualitative and quantitative perspectives, combining three assessment phases which are mentioned above: static identification, dynamic monitoring, and security assessment.

Models and methods: a weighted Hidden Markov Model (W-HMM) to establish the system’s security model with the algorithm of Baum-Welch is used.

Validation: an industrial robot system is used as an example; the effectiveness of DAF-ICSS is performed by comparing the two assessment methods.

Results: The simulation results show that the proposed DAF-ICSS is able to provide a more accurate assessment.

The advantages, limitations and ways to improve it are mentioned as follows:

The method can infer the risk value of the system through the explicit consequences of the attack on the device. The value can be used as system risk reference, but it cannot analyze the attack method.

However, this assessment method still has the room for improvement, such as how to select more reasonable weights, etc.

The proposed framework of DAF-ICSS is more sensitive than other methods.

The main findings are mentioned in conclusion as follow:

A hierarchical system is provided for evaluating security risk using the assessment method based on the W-HMM.

The degree of influence with different parameters in the factory is estimated to find out change of risk with high precision and efficiency

Recommendations:

Use the Sensors template available at https://www.mdpi.com/journal/sensors/instructions

The novelty and main contribution must be detailed at the end of introduction as well.

The main findings (qualitative and quantitative appreciations) must be better mentioned in conclusion (see below).

Include a discussion section where the obtained results, advantages, limitations and ways to improve the proposal are discussed and compared with other proposals proposed in the literature.

References

Avoid lumped references; see for example [4-7], [27-30] etc.; include a critical comment or appreciation for each reference or maximum two.

The proposed framework of DAF-ICSS is more sensitive than other methods [include at least one reference], which could also provide a better indication of dynamic change.

Minor editing errors; see for example:

Line 11: In the second stage, We propose …

Line 75: Industrial robots are playing an increasingly important role in the industrial field. [30].

Etc.

Revise the paper carefully.

Author Response

(The authors gave the same response as above.)

Round 2

Reviewer 2 Report

No further comments from my side

 The paper can be accepted in current form